METHODS

# A machine-learning method for biobank-scale genetic prediction of blood group antigens

Kati Hyvärinen[1]*, Katri Haimila[2], Camous Moslemi[3,4], Blood Service Biobank[5], Martin L. Olsson[6,7], Sisse R. Ostrowski[8,9], Ole B. Pedersen[3,9], Christian Erikstrup[10], Jukka Partanen[1], Jarmo Ritari[1]

1 Research and Development, Finnish Red Cross Blood Service, Helsinki, Finland, 2 Blood Group Unit, Finnish Red Cross Blood Service, Vantaa, Finland, 3 Department of Clinical Immunology, Zealand University Hospital, Køge, Denmark, 4 Department of Clinical Immunology, Aarhus University Hospital, Aarhus, Denmark, 5 Finnish Red Cross Blood Service, Vantaa, Finland, 6 Department of Laboratory Medicine, Lund University, Lund, Sweden, 7 Department of Clinical Immunology and Transfusion Medicine, Office for Medical Services, Region Skåne, Sweden, 8 Department of Clinical Immunology, Copenhagen University Hospital, Rigshospitalet, Copenhagen, Denmark, 9 Department of Clinical Medicine, University of Copenhagen, Copenhagen, Denmark, 10 Department of Clinical Immunology, Aarhus University Hospital, Skejby, Denmark

* kati.hyvarinen@bloodservice.fi

**Data Availability Statement:** Data contains potentially identifying sensitive information and cannot be publicly shared for ethical reasons. Genotyping and RBC antigen/phenotype and HPA-1 typing data for the Finnish cohort are stored in

## Abstract

A key element for successful blood transfusion is compatibility of the patient and donor red blood cell (RBC) antigens. Precise antigen matching reduces the risk for immunization and other adverse transfusion outcomes. RBC antigens are encoded by specific genes, which allows developing computational methods for determining antigens from genomic data. We describe here a classification method for determining RBC antigens from genotyping array data. Random forest models for 39 RBC antigens in 14 blood group systems and for human platelet antigen (HPA)-1 were trained and tested using genotype and RBC antigen and HPA-1 typing data available for 1,192 blood donors in the Finnish Blood Service Biobank. The algorithm and models were further evaluated using a validation cohort of 111,667 Danish blood donors. In the Finnish test data set, the median (interquartile range [IQR]) balanced accuracy for 39 models was 99.9 (98.9–100)%. We were able to replicate 34 out of 39 Finnish models in the Danish cohort and the median (IQR) balanced accuracy for classifications was 97.1 (90.1–99.4)%. When applying models trained with the Danish cohort, the median (IQR) balanced accuracy for the 40 Danish models in the Danish test data set was 99.3 (95.1–99.8)%. The RBC antigen and HPA-1 prediction models demonstrated high overall accuracies suitable for probabilistic determination of blood groups and HPA-1 at biobank-scale. Furthermore, population-specific training cohort increased the accuracies of the models. This stand-alone and freely available method is applicable for research and screening for antigen-negative blood donors.

the Blood Service Biobank, Helsinki, Finland. Researchers may apply for access to data from the Blood Service Biobank (https://www.veripalvelu.fi/en/biobank/for-researchers/; biopankki@veripalvelu.fi). Due to privacy laws, the Danish genetic data and phenotypes are only available to DBDS researchers and blood banks. DBDS has an established application process, which allows external researchers to get access to the DBDS biobank material (https://bloddonor.dk/bloddonorstudiet/the-danish-blood-donor-study-eng/; info@DBDS.dk). Code and model availability: Analysis scripts and the Finnish models are available at https://github.com/FRCBS/Blood_group_prediction.

**Funding:** The study was partially supported by funding from the Government of Finland VTR funding (to FRCBS, https://www.hus.fi/tutkimus-ja-opetus/tutkijan-ohjeet/valtion-tutkimusrahan-haku). The study of the Danish cohort was supported by Independent Research Fund Denmark (project number 0214-00127B, https://dff.dk/en), Bloddonorernes Forskningsfond (to CM, https://bloddonor.dk/info/om/fonde/), and A.P Møller Fonden (to CM, https://www.apmollerfonde.dk/). MLO is a Wallenberg Clinical Scholar funded by Knut and Alice Wallenberg Foundation (https://kaw.wallenberg.org/en). The funders had no role in study design, data collection and analysis, decision to publish, or preparation of the manuscript.

**Competing interests:** I have read the journal's policy and the authors of this manuscript have the following competing interests: M.L.O. is an inventor on patents about Vel blood group genotyping (unrelated to the methods and models presented in this study) and owns 50% each of the shares in BLUsang AB, an incorporated consulting firm, which receives royalties for said patents. The other authors declare no conflicts of interest.

## Author summary

Blood transfusion is one of the most common clinical procedures in the hospitals and the key element for safe transfusion is compatibility between the recipient and donor red blood cell antigens. Precise antigen matching reduces the risk for sensitization and other adverse transfusion outcomes. Here we describe a stand-alone and freely available random forest classification method and models for determining red blood cell and human platelet antigens from array technology-based genotyping data. We investigate the performance of models trained with Finnish blood donor biobank data and further validate the method with a large Danish cohort. The results demonstrate high overall accuracy, and the method is suitable for biobank-scale research and screening of antigen-negative donors. The implementation is possible in the local computing environment without sensitive data uploads and requires only a moderate level of bioinformatic skills.

## Introduction

Blood transfusion is a life-saving procedure performed widely in treating various medical conditions. Despite routine practices, the safety of transfusions remains a major concern [1]. Exposure to foreign RBC antigens may result in alloantibody formation and hemolytic transfusion reactions. Additionally, sensitization to non-self RBC antigens and human platelet antigens (HPAs) can also occur via pregnancy and cause fetal morbidity and mortality [2,3]. The current general practice of matching the recipient and blood donor for ABO and RhD antigens is inadequate to prevent sensitization to other antigens. Extended matching could reduce the risk of alloimmunization and adverse events, which are especially pronounced among patients receiving regular transfusions [4,5].

Blood group typing of blood donors has been conventionally performed by serotyping and is still the main method used in blood centers. To overcome limitations regarding low throughput and lack of valid reagents for all clinically relevant antigens, numerous DNA-based genotyping and sequencing methods have emerged within the last decades [6–10]. This development has been enabled by the accumulating knowledge about the genetic basis of the blood groups [11,12] and the rapid evolution of molecular methodology. However, the systematically extended blood group typing of blood donors and, even more so, the recipients, remains sparse. Economic feasibility has been a major restraint to the progress. The development of genotyping array technologies has promoted high-throughput and cost-effective genetic studies in many fields and, in 2020, Gleadall *et al.* [13] introduced a microarray platform for RBC antigen, human leukocyte antigens (HLA), and HPA typing for precision matching of blood.

While accurate blood group typing is obligatory for safe transfusions, an initial screening for potential donors could be achieved using less stringent procedures. In the last decade, the development of machine learning approaches for high-dimensional data has provided new opportunities for exploitation of expanding genetic data. In 2015, Giollo *et al.* [14] presented BOOGIE, an RBC antigen predictor based on Boolean rules and *k*-nearest neighbor (*k*-NN) algorithm. Decision tree -based methods, including bootstrap aggregation [15] and random forest [16], have been utilized for imputation of HLA alleles [17,18] and killer cell immunoglobulin-like receptor (KIR) copy number [19] and gene content [20]. To our knowledge, these methods have not yet been implemented on RBC antigen and HPA screening. The analysis of high-dimensional data with computational performance suitable for large-scale analyses may be implemented using "RANdom forest GEneRator" software R package [21]. The

execution is feasible in the local computing environment and sensitive data uploads are not required.

Here we describe a stand-alone and freely available random forest classification method and models for determining RBC antigens and HPA-1 from array technology-based genotyping data. We investigate the performance of models trained with Finnish blood donor biobank data and further validate the method with a Danish cohort. Our results suggest that the method is applicable for biobank-scale probabilistic determination of RBC antigens and HPA-1, and could facilitate research and screening for antigen-negative blood donors.

## Results

### Evaluation of the Finnish classification models

An overview of the study design is depicted in **Fig 1**. In the Finnish cohort, the genotype data was accessible for 1,192 blood donors and the RBC antigen typing data was available for 39 antigens representing 15 blood group systems. The blood group typing frequency varied greatly depending on RBC antigen/phenotype, being at the lowest 5% for HPA-1b and at the highest 100% for A, B, AB, O, K, D, C, c, E, and e (**Table 1**).

After data partitioning, the number of study subjects in the test data set was 596. The median (interquartile range [IQR]) balanced accuracy for 39 models was 99.9 (98.9–100)% in the test data set and accuracy metrics for all models are presented in **Table A in S1 Text**. The models for antigen/phenotype positivity of AB, B, $A_1$, $A_2$, $Yt^b$, $Co^a$, $Do^a$, $Do^b$, $Fy^a$, HPA-1b, K, $Kp^a$, $Ul^a$, $Jk^a$, $Lu^a$, S, and s reached balanced accuracy of 100%. For other models, the balanced accuracy was ≥98.0%, except 83.3% for $Ls^a$, 94.0% for $Le^b$, 95.0% for HPA-1a, and 96.0% for $hr^S$. Accuracy metrics for the train and full data sets are presented in **Tables B and C in S1 Text**, respectively. **Fig 2** illustrates the distributions of accuracy metrics over all antigens shared by different data sets. The number of false negative plus false positive (FN + FP) samples out of all samples was low, ranging from 0 to 1% in all models, except 2% for $hr^S$. Detailed confusion matrices for the Finnish test, train, and full data sets are presented in **Figs A–C in S1 Text**, respectively. The median (IQR) prediction error, determined as misclassification frequency obtained from out-of-bag data, of the Finnish models was $1.6 \times 10^{-3}$ ($1.9 \times 10^{-4}$–$7.0 \times 10^{-3}$) (**Table 2**). Receiver operating characteristic (ROC) and precision-recall curves for combined test data predictions are presented in **Fig D in S1 Text**. The area under ROC curve was 99.9% with confidence interval 99.8–99.9%.

The distributions of posterior probabilities (PP) in the test data set are depicted in **Fig E in S1 Text**. The samples having PP >0.5 were classified as antigen positive and ≤0.5 as antigen negative. The majority of the PPs were close to 1 for the antigen typing positive samples and close to 0 for the antigen typing negative samples. The $Co^a$-negative samples (only two samples in the test data set) were classified correctly but the PPs were closer to 0.5 than to 0. One of the three $Ls^a$-positive samples were misclassified and the PPs for the other two were closer to 0.5 than to 1 (specificity 66.7%). The spectrum of PP distribution with some misclassifications was observed for $Co^b$, $Le^b$, M, N, C, $C^w$, D, and $hr^S$. **Figs F and G in S1 Text** depict the distributions of PPs in the train and full data sets.

### Evaluation of prediction using gradient boosting method

We investigated the impact of machine learning algorithm selection using gradient boosting method. Combined accuracy metrics over all antigens shared by different data sets for the Finnish test data set for random forest and gradient boosting methods are presented in **Fig 2A and 2B**, respectively. **Table D in S1 Text** presents the detailed accuracy metrics, **Fig H in**

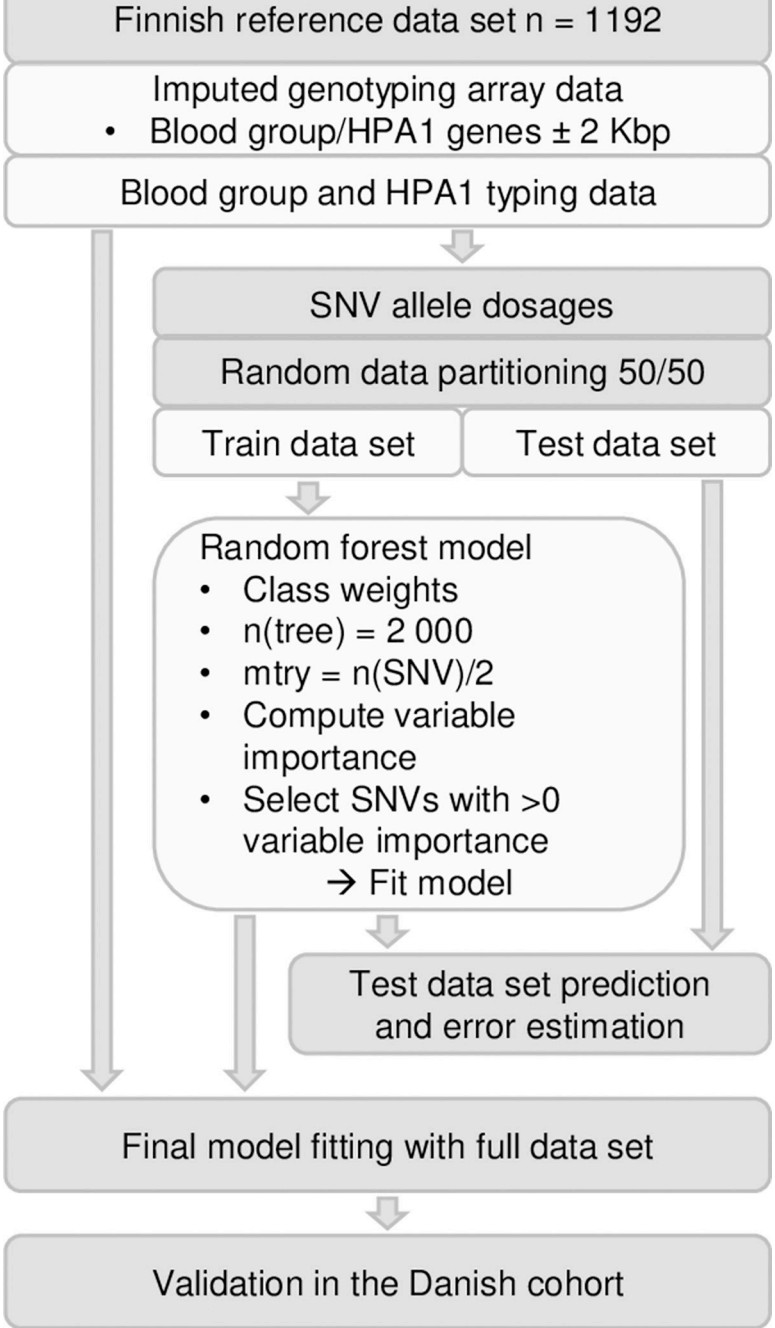

**Fig 1. Study design.** Random forest classification models were generated using Finnish reference data set (n = 1,192). Allele dosages of genes determining RBC antigens/phenotypes and HPA-1 were combined with the antigen typing data. The dataset was divided randomly to train and test data sets. Random forest modelling was executed in the training data set (n = 596) and the important variables were selected using permutation. The models were evaluated in the test data set (n = 596) for prediction accuracy and errors. The final models were fitted using the full data set and both models and the method were validated in the Danish cohort (n = 111,677).

**S1 Text** the confusion matrices, and **Fig I in S1 Text** the distributions of PPs in the Finnish test data set. The overall performance of gradient boosting was slightly lower than our random forest classification approach.

**Table 1. Blood group/HPA-1 antigen typing information of the Finnish and Danish cohorts.**

| Blood group/HPA system | Antigen[a] | Finnish cohort (n) | Antigen positivity in the Finnish cohort (%) | Danish cohort (n) | Antigen positivity in the Danish cohort (%) |
|---|---|---|---|---|---|
| ABO | A | 1,192 | 40.9 | 111,656 | 41.2 |
| ABO | $A_1$ | 70 | 62.9 | 17,541 | 75.3 |
| ABO | $A_2$ | 70 | 38.6 | 433 | 27.0 |
| ABO | AB | 1,192 | 8.0 | 111,656 | 4.4 |
| ABO | B | 1,192 | 8.2 | 111,656 | 10.2 |
| ABO | O | 1,192 | 42.9 | 111,656 | 44.2 |
| Cartwright | $Yt^a$ | NA | NA | 7,640 | 99.6 |
| Cartwright | $Yt^b$ | 1,160 | 6.2 | 7,362 | 7.3 |
| Colton | $Co^a$ | 1,162 | 99.8 | 7,490 | 98.5 |
| Colton | $Co^b$ | 1,164 | 7.3 | 8,277 | 9.2 |
| Dombrock | $Do^a$ | 1,162 | 53.1 | 7,352 | 62.9 |
| Dombrock | $Do^b$ | 1,162 | 90.7 | 7,352 | 84.6 |
| Duffy | $Fy^a$ | 1,177 | 67.0 | 77,920 | 67.0 |
| Duffy | $Fy^b$ | 1,175 | 75.1 | 71,096 | 80.8 |
| Gerbich | $Ls^a$ | 190 | 3.2 | NA | NA |
| HPA-1 | HPA-1a | 232 | 91.4 | 518 | 97.1 |
| HPA-1 | HPA-1b | 62 | 43.5 | 518 | 30.7 |
| Kell | K | 1,192 | 4.7 | 96,497 | 7.8 |
| Kell | k | NA | NA | 14,332 | 99.0 |
| Kell | $Kp^a$ | 1,177 | 2.6 | 22,497 | 2.4 |
| Kell | $Kp^b$ | NA | NA | 10,350 | 99.9 |
| Kell | $Ul^a$ | 219 | 15.5 | NA | NA |
| Kidd | $Jk^a$ | 1,177 | 72.6 | 77,986 | 76.6 |
| Kidd | $Jk^b$ | 1,177 | 68.5 | 71,840 | 72.7 |
| Knops | $Kn^a$ | NA | NA | 3,462 | 99.9 |
| Knops | $Kn^b$ | NA | NA | 3,462 | 6.8 |
| Landsteiner-Wiener | $LW^b$ | 327 | 5.2 | NA | NA |
| Lewis | $Le^a$ | 253 | 10.2 | 9,220 | 17.4 |
| Lewis | $Le^b$ | 253 | 80.2 | 8,925 | 56.5 |
| Lutheran | $Lu^a$ | 1,164 | 3.6 | 9,615 | 8.6 |
| MNS | M | 1,177 | 87.7 | 29,642 | 78.1 |
| MNS | N | 1,177 | 59.0 | 14,837 | 71.6 |
| MNS | S | 1,177 | 54.2 | 28,949 | 50.4 |
| MNS | s | 1,177 | 87.6 | 24,291 | 90.9 |
| P1PK | P1 | 253 | 75.9 | 7,465 | 77.4 |
| Rh | C | 1,192 | 54.4 | 44,451 | 63.3 |
| Rh | c | 1,192 | 79.4 | 42,968 | 82.2 |
| Rh | $C^W$ | 1,177 | 3.2 | 40,720 | 3.3 |
| Rh | $C^x$ | 337 | 12.2 | NA | NA |
| Rh | D | 1,192 | 71.6 | 111,667 | 79.5 |
| Rh | E | 1,192 | 19.9 | 89,289 | 28.1 |
| Rh | e | 1,192 | 96.2 | 82,467 | 97.3 |
| Rh | $hr^B$ | 1,162 | 96.4 | NA | NA |
| Rh | $hr^S$ | 1,162 | 97.3 | NA | NA |
| Vel | Vel | NA | NA | 11,755 | 99.9 |

NA, data not available.

[a] O, $A_1$, and $A_2$ and in this column refer to phenotype.

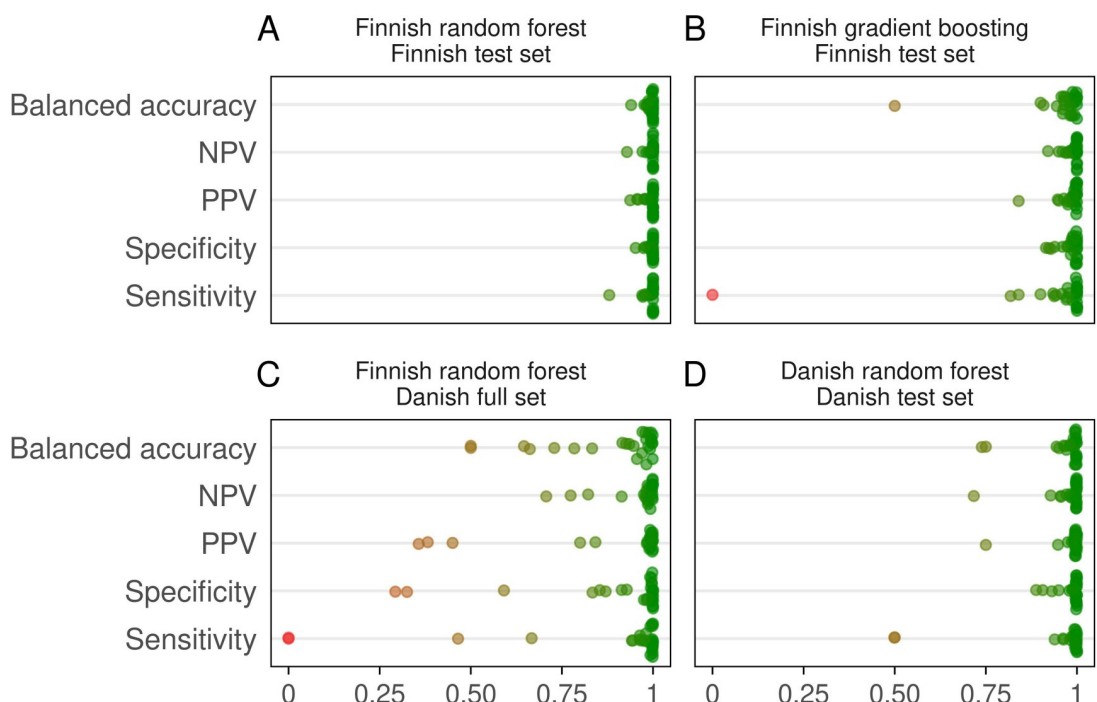

**Fig 2. Summary of prediction accuracy metrics.** Distributions of accuracy metrics over all antigens shared by different test data sets. (A) Finnish random forest models evaluated in the Finnish test data set. (B) Finnish gradient boosting models evaluated in the Finnish test data set. (C) Finnish random forest models evaluated in the Danish full data set. (D) Danish random forest models evaluated in the Danish test data set. NPV, negative predictive value; PPV, positive predictive value.

## Validation of the Finnish random forest classification models in the Danish cohort

The Danish validation cohort had genotype and phenotype data for 34 out of the 39 Finnish classification models. Antigen/phenotype typing data varied from 433 for $A_2$ to ~111,000 for A, AB, B, O, and D (**Table 1**). Due to missing Finnish model variables in the Danish genotype data, the Danish allele dosage data was harmonized using mean imputation before applying the Finnish models.

The median (IQR) balanced accuracy for classifications was 97.1 (90.1–99.4)% and all the evaluation metrics are presented in **Table E in S1 Text**. The balanced accuracies were >98.0% for 14 models including antigen/phenotype positivity of A, AB, B, O, Yt[b], Do[a], Do[b], HPA-1a, Jk[a], Le[a], S, s, E, and e. Models for antigen/phenotype positivity of $A_1$, Co[b], Fy[a], Fy[b], HPA-1b, K, Kp[a], Lu[a], M, N, and C[w] had balanced accuracy ranging from 91.6 to 98.0%. Six models, $A_2$, Co[a], Jk[b], Le[b], D, and C had balanced accuracy ranging from 64.6 to 89.4%. The Finnish models for LW[b], P1, and c failed classification in the Danish cohort. **Fig 2C** illustrates the distributions of accuracy metrics over all antigens shared by different data sets for Finnish random forest models evaluated in the Danish full data set.

## Validation of the random forest classification model algorithm in the Danish cohort

The RBC antigen/phenotype and HPA-1 typing and genotype data available for the Danish cohort enabled implementation of 40 Danish classification models representing 15 blood

**Table 2. Characteristics of the Finnish classification models.**

| Blood group/HPA system | Antigen[a] | Genes analyzed | n(variants available) | n(model variants) | Prediction error[b] |
|---|---|---|---|---|---|
| ABO | ABO | *ABO* | 496 | 164 | 8.68E-04 |
| ABO | $A_1$ | *ABO* | 496 | 18 | 1.29E-03 |
| ABO | $A_2$ | *ABO* | 496 | 43 | 2.20E-02 |
| Cartwright | Yt$^b$ | *ACHE* | 62 | 12 | 0.00E+00 |
| Colton | Co$^a$ | *AQP1* | 108 | 15 | 4.26E-04 |
| Colton | Co$^b$ | *AQP1* | 108 | 39 | 2.03E-03 |
| Dombrock | Do$^a$ | *ART4* | 109 | 17 | 6.97E-05 |
| Dombrock | Do$^b$ | *ART4* | 109 | 17 | 0.00E+00 |
| Duffy | Fy$^a$ | *ACKR1* | 57 | 25 | 6.80E-05 |
| Duffy | Fy$^b$ | *ACKR1* | 57 | 26 | 4.32E-03 |
| Gerbich | Ls$^a$ | *GYPC* | 342 | 81 | 7.71E-03 |
| HPA-1 | HPA-1a | *ITGB3* | 386 | 24 | 1.74E-03 |
| HPA-1 | HPA-1b | *ITGB3* | 386 | 73 | 2.06E-02 |
| Kell | K | *KEL* | 127 | 33 | 2.93E-05 |
| Kell | Kp$^a$ | *KEL* | 127 | 26 | 8.56E-05 |
| Kell | Ul$^a$ | *KEL* | 127 | 21 | 2.01E-04 |
| Kidd | Jk$^a$ | *SLC14A1* | 622 | 18 | 1.62E-04 |
| Kidd | Jk$^b$ | *SLC14A1* | 622 | 178 | 8.14E-04 |
| Landsteiner-Wiener | Lw$^b$ | *ICAM4* | 35 | 19 | 4.27E-03 |
| Lewis | Le$^a$ | *FUT2, FUT3* | 199 | 45 | 4.13E-03 |
| Lewis | Le$^b$ | *FUT2, FUT3* | 199 | 88 | 1.56E-02 |
| Lutheran | Lu$^a$ | *BCAM* | 124 | 14 | 9.47E-04 |
| MNS | M | *GYPA, GYPB, GYPE* | 688 | 132 | 6.80E-03 |
| MNS | N | *GYPA, GYPB, GYPE* | 688 | 214 | 1.18E-02 |
| MNS | S | *GYPA, GYPB, GYPE* | 688 | 44 | 1.54E-05 |
| MNS | s | *GYPA, GYPB, GYPE* | 688 | 49 | 1.19E-04 |
| P1PK | P1 | *A4GALT, B3GALNT1* | 368 | 135 | 1.73E-02 |
| Rh | C | *RHCE, RHD* | 266 | 108 | 9.23E-03 |
| Rh | c | *RHCE, RHD* | 266 | 68 | 3.23E-03 |
| Rh | C$^W$ | *RHCE, RHD* | 266 | 53 | 2.80E-03 |
| Rh | C$^x$ | *RHCE, RHD* | 266 | 12 | 4.70E-03 |
| Rh | D | *RHCE, RHD* | 266 | 113 | 7.47E-03 |
| Rh | E | *RHCE, RHD* | 266 | 80 | 1.07E-03 |
| Rh | e | *RHCE, RHD* | 266 | 23 | 1.33E-03 |
| Rh | hr$^B$ | *RHCE, RHD* | 266 | 20 | 1.39E-03 |
| Rh | hr$^S$ | *RHCE, RHD* | 266 | 22 | 7.71E-03 |

[a] O, $A_1$, and $A_2$ and in this column refer to phenotype.

[b] Misclassification frequency obtained from out-of-bag data.

group systems. Due to missing genotypes (approximately 5%), missing allele dosage values were imputed separately for train and test data sets using mean values.

Median (IQR) balanced accuracy for the 40 Danish models in the Danish test data set was 99.3 (95.1–99.8)%. The evaluation metrics for test data set are available in **Table F in S1 Text** and for the train and full data sets in **Tables G and H in S1 Text**, respectively. More than half (23/40) of the Danish models reached balanced accuracy of ≥99.0% including models for antigen/phenotype positivity of A, AB, B, O, Yt$^a$, Yt$^b$, Do$^a$, Do$^b$, Fy$^a$, HPA-1a, HPA-1b, Jk$^a$, Jk$^b$, M,

N, S, s, C, c, D, E, Le$^a$, and Kn$^b$. Balanced accuracies for A$_1$, Co$^b$, Fy$^b$, K, Kp$^a$, Lu$^a$, C$^w$, e, and P1 models ranged from 94.4 to 98.1%, and for A$_2$, Co$^a$, k, Kp$^b$, Lu$^b$, Vel, and Le$^b$ from 70.0 to 89.3%. Danish model for Kn$^a$ failed classification due to too low number of Kn$^a$-negative samples in the test data set. Confusion matrices for the Danish models in the Danish train, test and full data sets depict the distribution of true negative (TN), FN, true positive (TP), and FP samples and are illustrated in **Figs J–L in S1 Text**, respectively. The median (IQR) prediction error of the Danish models was 2.3 x 10$^{-3}$ (9.3 x 10−4–7.1 x 10$^{-3}$)% (**Table I in S1 Text**). The distributions of accuracy metrics over all antigens shared by different data sets for Danish random forest models evaluated in the Danish test data set are illustrated in **Fig 2D**.

### Comparison of the Finnish and Danish random forest classification models

Assembly of the balanced accuracies for Finnish and Danish models in the Finnish and Danish full data sets is presented in **Table 3**. When analyzing the shared 33 models, the Finnish models predicted the blood groups of the Finnish cohort more accurately than the blood groups of the Danish cohort (median [IQR] balanced accuracy 99.9 [98.8–100]% vs. 97.1 [91.6–99.5], p = 1.15e-06). The Danish models were performing better than the Finnish models in the blood group classification of the Danish cohort (median [IQR] balanced accuracy 99.5 [96.5–99.8]% vs. 97.1 [91.6–99.5]%, p = 0.006).

The number of genetic variants available for the Finnish random forest modelling ranged from 35 to 688 depending on the blood group/HPA system and number of the important variables selected by the classifier for the final models ranged from 12 to 214 (**Table 2**). In the Danish genotyping data set, the number of variants varied from 42 to 766 and the final models utilized 20–743 variants (**Table I in S1 Text**).

## Discussion

Our study introduces random forest classification models for predicting RBC antigens/phenotypes and HPA-1 from array-based genotyping data. The method and models were generated utilizing blood group typing data from Finnish blood donors and further validated using a large Danish blood donor cohort. The results demonstrate high overall accuracy, and the method is suitable for biobank-scale screening and analysis of HPA-1 and RBC antigens.

Blood transfusion is one of the most common clinical procedures in the hospitals and the key element for safe transfusion is compatibility between the recipient and donor RBC antigens [1]. Although transfusion-related severe outcomes are rare, the prominent risk of sensitization and further alloimmunization affects especially patients dependent on recurrent transfusions [4,5]. Extended blood group typing has proven to be beneficial by reducing the incidence of alloantibody formation [22,23]. Additionally, studies have shown that the extended genotyping of blood donors markedly increases the number of suitable donors for immunized recipients [13] and enhances the supply of antigen-negative blood [6].

At present, preventive matching strategies are implemented only for specific patient groups and, despite the obvious advantages of the extended genotyping of donors, the procedure has not been considered feasible covering all blood donors. Over the last decades, the genotyping of different populations has expanded widely. Using machine learning approaches to screen blood donor and research biobank genotyping data may provide a cost-effective solution for enlarging the pool of antigen-negative blood donors. Our random forest classification method infers RBC antigens and HPA-1 from genotype-imputed microarray data. The R package ranger performed fast and handled the dimensionality of input data without problems [21]. The obtained results demonstrated high balanced accuracies both in the Finnish discovery cohort (median 99.8% for the 39 Finnish models) and in the Danish validation cohort (median

**Table 3. Balanced accuracies for the Finnish and Danish models in full data sets.**

| Blood group/HPA system | Antigen[a] | Balanced accuracy | | |
|---|---|---|---|---|
| | | Finnish full data set Finnish models | Danish full data set Finnish models | Danish full data set Danish models |
| ABO | A | 0.999 | 0.995 | 0.999 |
| ABO | $A_1$ | 1.000 | 0.980 | 0.984 |
| ABO | $A_2$ | 0.981 | 0.894 | 0.891 |
| ABO | AB | 1.000 | 0.998 | 0.999 |
| ABO | B | 1.000 | 0.987 | 0.999 |
| ABO | O | 0.999 | 0.995 | 0.998 |
| Cartwright | Yt[a] | NA | NA | 1.000 |
| Cartwright | Yt[b] | 1.000 | 0.998 | 0.998 |
| Colton | Co[a] | 1.000 | 0.833 | 0.833 |
| Colton | Co[b] | 0.988 | 0.926 | 0.963 |
| Dombrock | Do[a] | 1.000 | 0.999 | 0.999 |
| Dombrock | Do[b] | 1.000 | 0.999 | 0.999 |
| Duffy | Fy[a] | 1.000 | 0.979 | 0.997 |
| Duffy | Fy[b] | 0.993 | 0.971 | 0.980 |
| Gerbich | Ls[a] | 0.917 | NA | NA |
| HPA-1 | HPA-1a | 1.000 | 1.000 | 1.000 |
| HPA-1 | HPA-1b | 0.986 | 0.939 | 0.999 |
| Kell | K | 1.000 | 0.916 | 0.956 |
| Kell | k | NA | NA | 0.838 |
| Kell | Kp[a] | 1.000 | 0.935 | 0.961 |
| Kell | Kp[b] | NA | NA | 0.750 |
| Kell | Ul[a] | 1.000 | NA | NA |
| Kidd | Jk[a] | 1.000 | 0.995 | 0.997 |
| Kidd | Jk[b] | 0.999 | 0.662 | 0.997 |
| Knops | Kn[a] | NA | NA | 0.750 |
| Knops | Kn[b] | NA | NA | 1.000 |
| Landsteiner_Wiener | LW[b] | 0.998 | 0.500 | NA |
| Lewis | Le[a] | 0.998 | 0.981 | 0.989 |
| Lewis | Le[b] | 0.968 | 0.729 | 0.736 |
| Lutheran | Lu[a] | 0.988 | 0.956 | 0.964 |
| Lutheran | Lu[b] | NA | NA | 0.828 |
| MNS | M | 0.984 | 0.970 | 0.993 |
| MNS | N | 0.988 | 0.947 | 0.987 |
| MNS | S | 1.000 | 0.995 | 0.997 |
| MNS | s | 1.000 | 0.988 | 0.993 |
| P1PK | P1 | 0.984 | 0.500 | 0.965 |
| Rh | C | 0.989 | 0.646 | 0.997 |
| Rh | c | 0.997 | 0.500 | 0.997 |
| Rh | C[W] | 0.999 | 0.918 | 0.953 |
| Rh | C[x] | 0.997 | NA | NA |
| Rh | D | 0.988 | 0.784 | 0.997 |
| Rh | E | 0.998 | 0.992 | 0.995 |
| Rh | e | 0.988 | 0.982 | 0.984 |
| Rh | hr[B] | 0.988 | NA | NA |
| Rh | hr[S] | 0.933 | NA | NA |

(*Continued*)

**Table 3.** (Continued)

| Blood group/HPA system | Antigen[a] | Balanced accuracy | | |
|---|---|---|---|---|
| | | Finnish full data set Finnish models | Danish full data set Finnish models | Danish full data set Danish models |
| Vel | Vel | NA | NA | 0.811 |

NA, data not available.

[a] O, $A_1$, and $A_2$ in this column refer to phenotype.

99.3% for the 40 Danish models) (**Table 3**). The performance was not affected by nearly a 100-fold size difference between the Finnish and the Danish cohorts (~1,200 vs. ~111,000, respectively).

Rh and MNS blood group system antigens have been challenging to determine by sequencing due to complex genetic variation and gene rearrangements [12,24]. We observed reduced balanced accuracy in the Finnish model for $hr^S$ (93.3%) and the Danish model for $C^w$ (95.3%). However, the other Rh and MNS antigen models, including clinically significant E, e, C, c, S, and s, performed accurately. The balanced accuracies for clinically significant antigens in other systems, including K, $Jk^a$, $Jk^b$, $Fy^a$, and $Fy^b$, ranged from 95.6% to 100% (**Table 3**).

The BOOGIE method for prediction of RBC antigens was published in 2015 [14]. It builds on 1-NN algorithm and implementation requires genotype sequencing data and curated haplotype tables for the RBC antigen phenotypes. When compared, the Finnish models for ABO and RhD performed better than the BOOGIE method (median balanced accuracy for the Finnish ABO models 100% vs. BOOGIE ABO accuracy 94.2%; balanced accuracy for the Finnish RhD model 98.8% vs. BOOGIE RhD accuracy 94.2%). The observed differences in accuracies could be explained by the potentially limited haplotype tables utilized by BOOGIE. Additionally, the reported results of BOOGIE are based on low number of samples.

When applying the Finnish models to the Danish cohort, the observed decrease in balanced accuracies was expected because of the evident genetic, genotyping, and imputation differences between the Finnish and the Danish cohorts (**Table 3**). The Finnish cohort was imputed using population-specific imputation reference panel having no missingness per individual. On the contrary, the Danish cohort was imputed using the North European reference sequence panel resulting in an average missingness of 5%. As random forest is not able to handle missing input data and the important variables of the Finnish models were not fully present in the Danish data, we were obliged to use mean imputation for missing variant dosage data. It is obvious that this approach also introduces errors to the data, which may partly explain the reduced accuracy. The better performance of the Danish models in Danish cohort underlined the benefit of the population-specific training cohort.

Tree-based ensemble methods such as random forest offer robust performance with low risk of overfitting and gradient boosting [25] may increase accuracy by modelling residuals. However, our XGBoost test controlling overfitting via cross-validation did not result in equal performance when compared to our random forest approach (**Fig 2**), suggesting that our model makes efficient use of available genetic data. Neural network models are a lucrative option for imputing missing data but parameter tuning benefits from large training data [26]. However, if the input data is relatively small, using specialized tools designed for imputing missing genotypes independently of input data size prior to modelling are a superior option to neural networks requiring large data sets. The most effective imputation of input genotypes is achieved using a population-specific reference panel [27]. In our case, we did not have large

training set at our disposal and had only limited data from rare antigen types available. Therefore, we adopted a modelling approach able to efficiently handle this kind of data.

The Finnish genotyping data had only one variant in the *RHD* region. Nonetheless, the Finnish model for RhD performed with sufficient balanced accuracy in the Finnish cohort (98.8%). Our method combines *RHD* and *RHCE* region variants for the modelling and the high linkage disequilibrium may have supported the classification (**Table A in S1 Text**). However, the Finnish model for RhD worked poorly in the Danish cohort (78.4%), which may be attributed to the mean imputation of missing values.

The present modelling method is restricted to the RBC antigen typing data available for the training and test data sets, which can be considered as a major limitation because the data for some RBC antigens are scarce. RBC antigens have demonstrated significant diversity among populations and rare blood group variants may not be discovered without substantially large typing numbers. The Danish model for Kn[a] failed because of lacking Kn[a]-negative samples in the test data set and we were not able to create Finnish models for e.g., Vel, k, Kp[b], Lu[a], and LW[a]. It would be beneficial to validate the present method and models in non-European populations to enable systematic blood group studies in biobanks of different ethnic origins and phenotype data content.

To our surprise, *B3GALNT1* on chromosome 3 supported the prediction of P1 antigen status in the P1PK system, even if this system is known to be governed by *A4GALT* on chromosome 22. *B3GALNT1* normally governs expression of the P and other antigens in the GLOB system [28]. Thus, our data may suggest an unknown but intriguing role of the glycosyltransferase encoded by *B3GALNT1* in the synthesis of P1 antigen. This deserves further investigation beyond the scope of this study.

In the future, comprehensive donor and recipient typing and precision matching are likely to increase. A recent publication by van Sambeeck *et al.* [29] demonstrated the feasibility of preventive matching for all genotyped recipients and donors. Our method is suitable for initial screening for antigen-negative donors at biobank-scale, presenting a cost-effective solution for the extended blood group and HPA-1 typing. Additionally, successful prediction of polygenic blood groups may facilitate the research of disease associations in large biobanks.

Scripts for random forest modelling and for applying the tested 39 Finnish models are freely available in the GitHub. The implementation is possible in the local computing environment without sensitive data uploads and requires only a moderate level of bioinformatic skills.

## Study subjects and methods

### Ethics statement

The Finnish study cohort consists of 1,192 blood donors belonging to the Blood Service Biobank, Helsinki, Finland (https://www.veripalvelu.fi/en/biobank/). Genotype and blood group phenotype data were obtained from the Blood Service Biobank. The study (biobank decision 002–2018) conforms to the principles of the Finnish Biobank Act (688/2012) and the participants have given written informed consent to the Blood Service Biobank.

The Danish validation cohort consists of 111,667 participants of the Danish Blood Donor Study (DBDS) Genomic Cohort expanding on the Danish blood bank system [30,31]. The genetic studies in DBDS have been approved by the Danish Data Protection Agency (P-2019-99) and the Scientific Ethical Committee system (NVK-1700407).

### Genotyping and genotype imputation

The genotyping and genotype imputation of the Finnish cohort have been performed originally as a part of FinnGen project (https://www.finngen.fi/en). Biobank samples were

genotyped using FinnGen ThermoFisher Axiom custom array v2 (Thermo Fisher Scientific, Santa Clara, CA, USA) and imputed using the population-specific Sisu v3 imputation reference panel with Beagle 4.1. Detailed description of the procedures is available at https://finngen.gitbook.io/documentation/v/r4/methods/genotype-imputation and the marker content of the custom array v2 is downloadable at https://www.finngen.fi/en/researchers/genotyping. The phased genotypes were filtered for the imputation INFO-score >0.6 and were in vcf format.

In the Danish cohort, the genotyping was performed using Illumina's Infinium Global Screening Array and imputed using the deCODE genetics' (Reykjavik, Iceland) North European reference sequence panel. Unphased genotypes were filtered for the imputation INFO-score >0.75, minor allele frequency >0.01, Hardy–Weinberg equilibrium P-values $<1 \times 10^{-4}$, and samples for missingness per individual <3%.

## RBC antigen and HPA typing

The RBC antigen and HPA-1 phenotypic information for the Finnish and Danish cohorts is presented in **Table 1**. The availability of the phenotype data varied in a wide range depending on the antigen due to the different testing criteria practices. In the Finnish cohort, RBC antigen and HPA-1 typing was performed at the FRCBS Blood Group Unit by routine methods and the results were obtained using validated serological and genotyping techniques.

The sources for RBC antigen and HPA-1 typing results were the Danish electronic blood bank systems and the typing was performed using serological methods, except for Vel-status, which was determined using polymerase chain reaction technique.

## Classification random forest models

**Fig 1** presents an overview of the study design. RBC antigen and HPA-1 coding genes and the genetic regions used in the models are presented in **Table J in S1 Text**. The input and output of the model fitting are presented in **Fig 3**. The input of the model is imputed genotype data in chromosomal variant call format (VCF) and antigen data in text format (e.g., Kpa+/Kpa-). The outputs are the models and associated information in R Data Serialization (RDS) format and accuracy statistics and important variables in figure and text format. The genomic regions of the genes encoding the target antigens are extracted from the VCF data, converted to PLINK format, and further into allele dosages. The models for the antigens were generated separately using the same hyperparameters. Only antigens having at least four cases in each respective typing data class were included, resulting altogether in 39 models. For the Finnish reference data set, SNVs in RBC antigen and HPA-1 coding genetic regions ± 2,000 bp flanking regions were utilized in dosage format. **Table 2** presents the number of SNVs available for each model. Only samples having full dosage data were used. The genetic and antigen typing information were combined into a single full data set and divided randomly 1:1 into train and test data sets.

R v4.3.0 environment [32] was used for the implementation of analyses. Classification random forest models were created using the R package ranger v0.13.1[21]. The number of trees was 2,000 and split criteria based on node impurity measured by the Gini index. Class weights were applied due to unbalanced outcome classes. Number of variables to possibly split at each node (mtry) was number of SNVs divided by 2 and the variable importance was determined by permutation. Feature selection was based on variable importance >0 and the model was re-fitted using these important SNVs only. The number of important variables and prediction errors for each antigen model are presented in the **Table 2**. Prediction error was determined as misclassification frequency obtained from out-of-bag data and prediction on the test set. The

INPUTS MODEL FITTING OUTPUTS

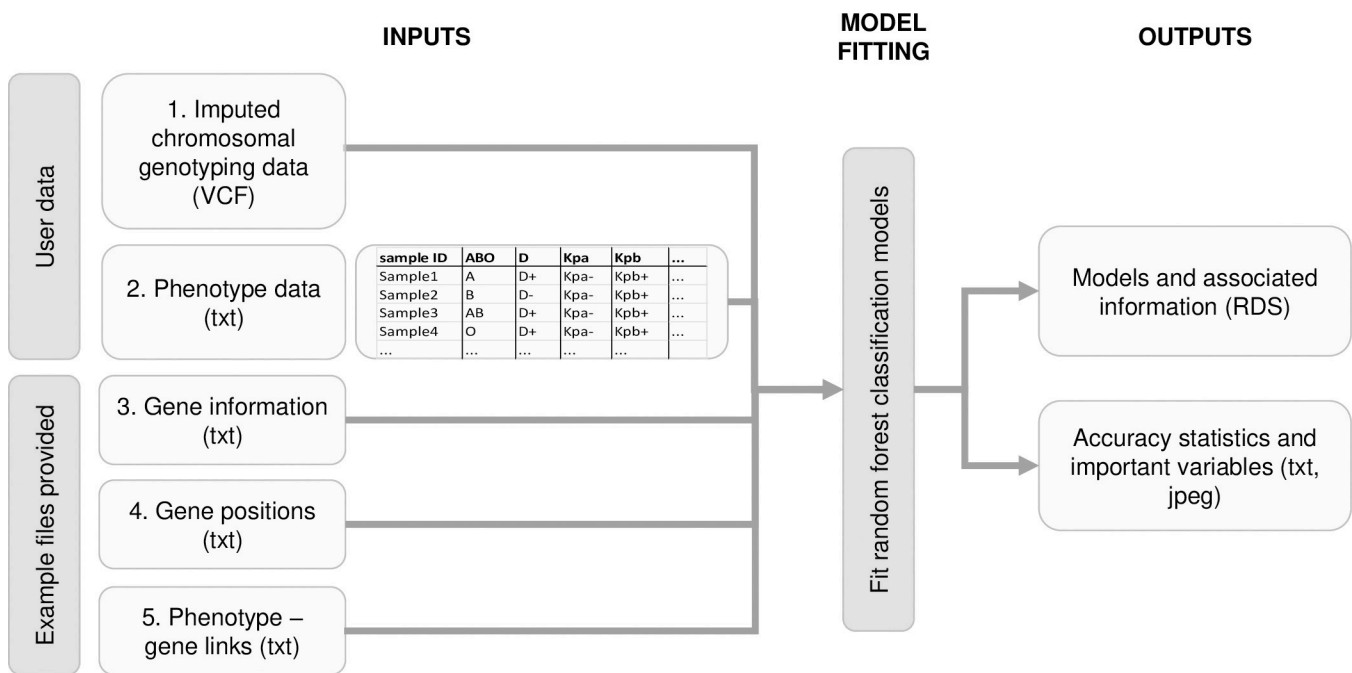

**Fig 3. Random forest model fitting.** Input data for model fitting include target genotype and phenotype data and gene-phenotype data provided in the GitHub repository https://github.com/FRCBS/Blood_group_prediction. Outputs of the classification are models for the target antigens and related accuracy information.

important variables and their importance values for the Finnish models are listed in **S1 Data**. The full data set was used in fitting the final models.

## Evaluation of prediction using gradient boosting method

To compare our random forest with feature selection approach to another tree-based classification algorithm, we fitted a binary logistic eXtreme Gradient Boosting (XGBoost)[25] model implemented by the R library xgboost v 1.7.6.1(33) to the Finnish training set data and evaluated its performance in the independent test set. To minimize overfitting of the XGBoost model, we performed 100 random data partitionings (2/3 train, 1/3 test) within the training set and selected the optimal number of boosting rounds based on minimum negative log-likelihood from each iteration. Within the iterations, we used an early_stopping_rounds parameter value of 4, indicating that the training with a validation set stops if the performance doesn't improve for four rounds. The final number of boosting rounds applied to the model fitted on the full training data was an average over the 100 iterations.

## Model evaluation metrics

The model accuracy was evaluated using sensitivity, specificity, positive predictive value (PPV), negative predictive value (NPV), and balanced accuracy. The data was wrangled using tidyverse v1.3.1 package[34] and the evaluation metrics were derived using caret v6.0–92[35]. For each model, the number of true positives (TP), true negatives (TN), false positives (FP) and false negatives (FN) were determined. Sensitivity was defined as TP / (TP + FN), specificity as TN / (TN + FP), PPV as TP / (TP + FP), NPV as TN / (TN + FN)[36]. Balanced accuracy accounts for imbalanced classification and was defined as (sensitivity + specificity) / 2. ROC and precision-recall curves were generated using ROCR v. 1.0–11 package.

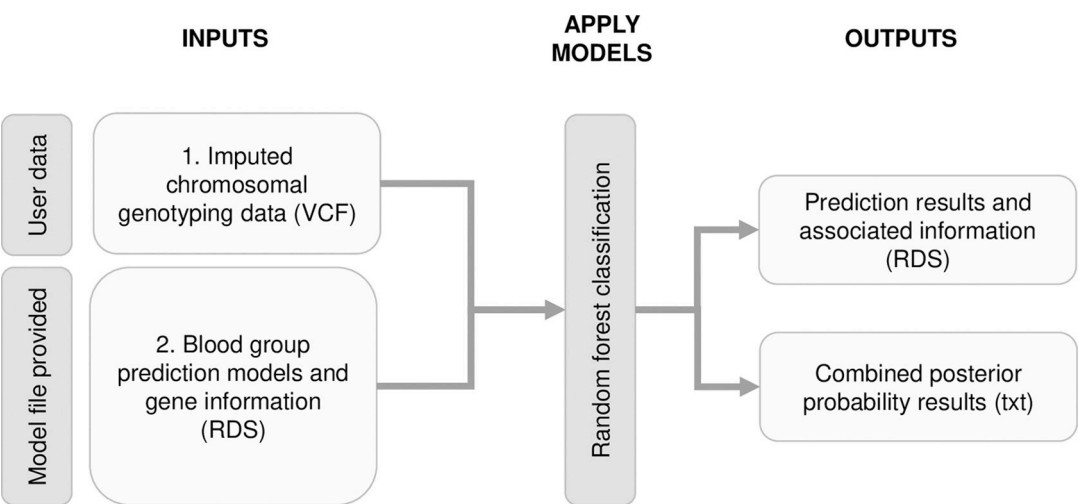

**Fig 4. Application of the Finnish models.** Input data for application include target genotype data and Finnish model file provided in the GitHub repository https://github.com/FRCBS/Blood_group_prediction. Outputs of the classification are prediction results for the antigens.

## Validation of the Finnish models and the random forest method for generating the models

The models obtained using the Finnish data set were applied to the Danish cohort. The input and output of the application of the Finnish models are presented in the **Fig 4**. The implementation required imputed genotype data in chromosomal variant call (or PLINK) format and provided Finnish prediction models and gene information in RDS format. The outputs were prediction results and associated information (RDS), and combined posterior probability results in text format. The Danish allele dosage data was harmonized by naming and allele orientation for compatibility with the Finnish models and the dosage data for the missing important variables was imputed using mean values.

The model-generating method was further validated by fitting the models on the Danish data set to create models specific to the Danish cohort. In the Danish data set, the percentage of missing genotypes was on average 5% depending on the genetic region of the blood group/HPA system. Missing allele dosage values were imputed separately for train and test data sets using mean values before classification random forest step. Characteristics of the Danish models are presented in **Table I in S1 Text**. The important variables and their importance values for the Danish models are listed in **S2 Data**. The evaluation metrics for both prediction and modelling were defined as depicted in the "*Model evaluation metrics*" section.

The significance of variation of balanced accuracies was analyzed using Mann-Whitney-Wilcoxon Test implemented with R v3.6.1.

## Supporting information

**S1 Text. Document contains supplementary tables and figures.** Table A. Accuracy metrics for the Finnish random forest models in the Finnish test data set. Table B. Accuracy metrics for the Finnish random forest models in the Finnish train data set. Table C. Accuracy metrics for the Finnish random forest models in the Finnish full data set. Table D. Accuracy metrics for the Finnish gradient boosting models in the Finnish test data set. Table E. Accuracy metrics for the Finnish random forest models in the Danish full data set. Table F. Accuracy metrics for the Danish random forest models in the Danish test data set. Table G. Accuracy metrics for the

Danish random forest models in the Danish train data set. Table H. Accuracy metrics for the Danish random forest models in the Danish full data set. Table I. Characteristics of the Danish random forest classification models. Table J. Blood group/HPA-1 genes and genetic regions. Fig A. Confusion matrices for the Finnish random forest models in the Finnish test data set. Fig B. Confusion matrices for the Finnish random forest models in the Finnish train data set. Fig C. Confusion matrices for the Finnish random forest models in the Finnish full data set. Fig D. Receiver operating characteristic and precision-recall curves for the Finnish random forest models in the Finnish test data se. Fig E. Posterior probability boxplots for the Finnish random forest models in the Finnish test data set. Fig F. Posterior probability boxplots for the Finnish random forest models in the Finnish train data set. Fig G. Posterior probability box-plots for the Finnish random forest models in the Finnish full data set. Fig H. Confusion matrices for the Finnish gradient boosting models in the Finnish test data set. Fig I. Posterior probability boxplots for the Finnish gradient boosting models in the Finnish test data set. Fig J. Confusion matrices for the Danish random forest models in the Danish train data set. Fig K. Confusion matrices for the Danish random forest models in the Danish test data set. Fig L. Confusion matrices for the Danish random forest models in the Danish full data set.
(PDF)

**S1 Data. Document contains list of important variables for the Finnish random forest models.**
(XLSX)

**S2 Data. Document contains list of important variables for the Danish random forest models.**
(XLSX)

## Acknowledgments

We want to thank Dr. Satu Pastila and Ms. Ritva Toivanen at the FRCBS for the collaboration with the blood group typing data. We are also grateful for Ms. Birgitta Rantala, Mr. Petteri Vaskin, Ms. Katariina Karjalainen, Ms. Nina Nikiforow, Ms. Jonna Clancy, and Dr. Mikko Arvas and Dr. Tiina Wahlfors at the Blood Service Biobank for their help in handling the data and samples, and Dr. Jaana Mättö and the personnel at the FRCBS Blood Group Unit for blood group typing analyses. From Denmark, we wish to thank the Danish blood donors and deCODE Genetics for genotyping the Danish cohort.

## Author Contributions

**Conceptualization:** Kati Hyvärinen, Katri Haimila, Jukka Partanen, Jarmo Ritari.

**Data curation:** Kati Hyvärinen, Katri Haimila, Camous Moslemi, Blood Service Biobank.

**Formal analysis:** Kati Hyvärinen, Camous Moslemi, Jarmo Ritari.

**Funding acquisition:** Camous Moslemi, Martin L. Olsson, Ole B. Pedersen, Christian Erikstrup, Jukka Partanen.

**Investigation:** Kati Hyvärinen, Katri Haimila, Camous Moslemi, Jarmo Ritari.

**Methodology:** Kati Hyvärinen, Camous Moslemi, Jarmo Ritari.

**Project administration:** Kati Hyvärinen, Ole B. Pedersen, Christian Erikstrup, Jukka Partanen.

**Resources:** Katri Haimila, Blood Service Biobank, Ole B. Pedersen, Christian Erikstrup, Jukka Partanen.

**Software:** Kati Hyvärinen, Camous Moslemi, Jarmo Ritari.

**Supervision:** Martin L. Olsson, Sisse R. Ostrowski, Ole B. Pedersen, Christian Erikstrup.

**Validation:** Kati Hyvärinen, Katri Haimila, Camous Moslemi, Jarmo Ritari.

**Visualization:** Kati Hyvärinen, Jarmo Ritari.

**Writing – original draft:** Kati Hyvärinen, Jarmo Ritari.

**Writing – review & editing:** Kati Hyvärinen, Katri Haimila, Camous Moslemi, Martin L. Olsson, Sisse R. Ostrowski, Ole B. Pedersen, Christian Erikstrup, Jukka Partanen, Jarmo Ritari.

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
