## [Decision Letter · Decision Letter 0]

6 Dec 2023

Dear Dr Hyvärinen,

Thank you very much for submitting your manuscript "A machine-learning method for biobank-scale genetic prediction of blood group antigens" for consideration at PLOS Computational Biology.

As with all papers reviewed by the journal, your manuscript was reviewed by members of the editorial board and by several independent reviewers. In light of the reviews (below this email), we would like to invite the resubmission of a significantly-revised version that takes into account the reviewers' comments.

We cannot make any decision about publication until we have seen the revised manuscript and your response to the reviewers' comments. Your revised manuscript is also likely to be sent to reviewers for further evaluation.

Sincerely,

Yang Lu, Ph.D.

Academic Editor

PLOS Computational Biology

Pedro Mendes

Section Editor

PLOS Computational Biology

Reviewer's Responses to Questions

**Comments to the Authors:**

Reviewer #1: This manuscript introduces a novel random forest classification approach, designed as a stand-alone and freely accessible tool, for the probabilistic determination of red blood cell (RBC) antigens and HPA-1 from genotyping data obtained via array technology. The results demonstrate high overall accuracy, and the method is suitable for biobank-scale research and screening of antigen-negative donors. In reviewing this manuscript, I appreciate its contributions to the field; however, I have several concerns that, if addressed, could strengthen the paper and its clarity:

1.For the computational model presented, it would be beneficial for the manuscript to articulate more clearly the model's inputs and outputs in the Methods section. A detailed description of what data the model requires and what results it produces will aid in understanding and potentially increase the manuscript's appeal to a broader audience.

2.While the random forest model has been emphasized, it would be valuable to see a comparison with other models, including different ensemble models or neural network models.

3.Given that variational neural network models have been shown to handle or impute missing values effectively in feature space, it would be interesting to explore or discuss whether such models could be applicable or advantageous in this context.

4.The manuscript could benefit from a more streamlined presentation of the figures in the main text. Reducing the number of sub-images or focusing on the most illustrative ones could enhance the clarity and conciseness of the paper.

Addressing these points would not only improve the clarity of the manuscript but also expand its depth and comparative context, potentially making it a more comprehensive and valuable resource for the field.

Reviewer #2: Overall, this paper does an excellent job. This paper effectively demonstrates the feasibility and high accuracy of using computational methods for determining RBC antigens and HPA-1 from genotyping array data. The rigorous testing and validation across Finnish and Danish cohorts underscore the robustness of the proposed models. Furthermore, the high balanced accuracy rates achieved highlight the potential of these models for large-scale applications in blood transfusion and genomics research. The paper's clear presentation of methodologies and findings significantly contributes to advancing precision medicine in the field of blood transfusion

I have a few comments:

1.Figure Resolution: The resolution of the figures is quite poor. Please verify if this is an issue with the review system.

2. Model Analysis: I recommend including confusion matrices for the Finnish model, validated using Danish cohorts.

3. Presentation of Accuracy Metrics: For tables like Table 2 and 4, consider using a boxplot or swarm plot for better clarity. Detailed tables could be moved to supplementary material to enhance readability.

4. Interpretation of Data (Line 160): Describing 23 out of 40 as a 'majority' may not be accurate. Please reconsider this characterization.

5. Model Interpretation: Given the impressive performance, employing explainable methods, such as feature importance or Shapley values, for interpreting the model is suggested.

Reviewer #3: In this manuscript, Hyvärinen et al. adopted random forest models for determining compatibility of the patient and donor red blood cell (RBC) antigens from genotyping array data. By testing on Finnish Blood Service and validating on a Danish blood cohort. The manuscript is organized and relatively easy to read.

I have 5 major concerns.

1 Can you add a schematic figure to explain the input and output of the model? It is not clear to me what is the dimension of the input genotype data and what is the representation of antigen. Is it just one-hot encoding?

2 Have the authors considered discussing the potential use cases of these models in other biobank data?

3 Are there some competing methods for the same tasks? Have the authors considered other ML models such as XGBoosting?

4 It would be more intuitive to show AUROC curve and precision-recall curve.

5 The visualization and table need to be greatly improved. All models seem performing reasonably well. A lot of results can be put into supplementary results. The authors need to think carefully how to summarize the results for the audience.

I have 1 minor comment.

The authors validate using the Danish cohort. Have the authors considered validating the Africa/Asia cohort?

**Have the authors made all data and (if applicable) computational code underlying the findings in their manuscript fully available?**

Reviewer #1: Yes

Reviewer #2: Yes

Reviewer #3: Yes

PLOS authors have the option to publish the peer review history of their article (what does this mean?). If published, this will include your full peer review and any attached files.

Reviewer #1: No

Reviewer #2: No

Reviewer #3: No
---

## [Decision Letter · Decision Letter 1]

7 Mar 2024

Dear Dr Hyvärinen,

We are pleased to inform you that your manuscript 'A machine-learning method for biobank-scale genetic prediction of blood group antigens' has been provisionally accepted for publication in PLOS Computational Biology.

Best regards,

Yang Lu, Ph.D.

Academic Editor

PLOS Computational Biology

Pedro Mendes

Section Editor

PLOS Computational Biology

Reviewer's Responses to Questions

**Comments to the Authors:**

Reviewer #1: The authors have solved all my concerns.

Reviewer #2: I believe the authors have addressed the issue I raised, so I would be happy to recommend acceptance

Reviewer #3: The authors have addressed most of my concerns. But I think the authors should still clarify:

1. When combining two networks, the authors fitted the calibration functions with KEGG pathway information, which makes this a supervised method. The authors should make this explicit for the results in Figures 3 and 4.

2. Have the authors considered other unsupervised approaches to combine two networks? In theory, it might not perform as well as the supervised method, but it should still outperform each network.

3. I think the authors should emphasize that the results in Figure 2 if I understand correctly, are invariant to the calibrating functions that FAVA fits.

4. How does the combination of two networks work exactly? Do you average/multiply two probability scores? I did not find these details in the manuscript.

**Have the authors made all data and (if applicable) computational code underlying the findings in their manuscript fully available?**

Reviewer #1: Yes

Reviewer #2: Yes

Reviewer #3: Yes

PLOS authors have the option to publish the peer review history of their article (what does this mean?). If published, this will include your full peer review and any attached files.

Reviewer #1: No

Reviewer #2: No

Reviewer #3: No

---

## [Editor Report · Acceptance letter]

18 Mar 2024

PCOMPBIOL-D-23-01617R1 

A machine-learning method for biobank-scale genetic prediction of blood group antigens

Dear Dr Hyvärinen,

I am pleased to inform you that your manuscript has been formally accepted for publication in PLOS Computational Biology. Your manuscript is now with our production department and you will be notified of the publication date in due course.

With kind regards,

Anita Estes
